Indole and 2,4-Thiazolidinedione conjugates as potential anticancer modulators

Corigliano Domenica M. 1
Syed Riyaz 2
Messineo Sebastiano 1
Lupia Antonio 1
Patel Rahul 3
Reddy Chittireddy Venkata Ramana 2
Dubey Pramod K. 2
Colica Carmela 4
Amato Rosario 1
De Sarro Giovambattista 1
Alcaro Stefano 1
Indrasena Adisherla adisherla.indrasena@gmail.com 2
Brunetti Antonio brunetti@unicz.it 1
1 Department of Health Sciences, University “Magna Græcia” of Catanzaro , Catanzaro , Italy
2 Department of Chemistry, Jawaharlal Nehru Technological University , Kukatpally, Hyderabad , India
3 Department of Food Science and Biotechnology, Dongguk University , Ilsandong-gu, Goyang-si, Gyeonggi-do , South Korea
4 CNR, IBFM UOS of Germaneto, University “Magna Græcia” of Catanzaro , Catanzaro , Italy
Singh Shree Ram
Electronic publication date: 2018 Aug 8
Publication date: 2018
Volume: 6
Electronic Location ID: e5386
Received 2018 Feb 15; Accepted 2018 Jul 17
Copyright: ©2018 Corigliano et al.
Copyright year: 2018
Copyright holder: Corigliano et al.
License: This is an open access article distributed under the terms of the Creative Commons Attribution License, which permits unrestricted use, distribution, reproduction and adaptation in any medium and for any purpose provided that it is properly attributed. For attribution, the original author(s), title, publication source (PeerJ) and either DOI or URL of the article must be cited.
License URL: https://creativecommons.org/licenses/by/4.0/

Keywords: Thiazolidinediones, Cancer, Cellular viability, Wound healing, Cell proliferation, BCL-xL, Apoptosis

Funding: Agenzia Italiana del Farmaco, AIFA, Regione Calabria Project This study was supported by a Grant from the Agenzia Italiana del Farmaco, AIFA, Regione Calabria Project “Rete Regionale di informazione sul farmaco: informazione, formazione, e farmacovigilanza”. There was no additional external funding received for this study. The funders had no role in study design, data collection and analysis, decision to publish, or preparation of the manuscript.

==============================
Background

Thiazolidinediones (TZDs), also called glitazones, are five-membered carbon ring molecules commonly used for the management of insulin resistance and type 2 diabetes. Recently, many prospective studies have also documented the impact of these compounds as anti-proliferative agents, though several negative side effects such as hepatotoxicity, water retention and cardiac issues have been reported. In this work, we synthesized twenty-six new TZD analogues where the thiazolidinone moiety is directly connected to an N-heterocyclic ring in order to lower their toxic effects.

Methods

By adopting a widely applicable synthetic method, twenty-six TZD derivatives were synthesized and tested for their antiproliferative activity in MTT and Wound healing assays with PC3 (prostate cancer) and MCF-7 (breast cancer) cells.

Results

Three compounds, out of twenty-six, significantly decreased cellular viability and migration, and these effects were even more pronounced when compared with rosiglitazone, a well-known member of the TZD class of antidiabetic agents. As revealed by Western blot analysis, part of this antiproliferative effect was supported by apoptosis studies evaluating BCL-xL and C-PARP protein expression.

Conclusion

Our data highlight the promising potential of these TZD derivatives as anti-proliferative agents for the treatment of prostate and breast cancer.

Introduction

Thiazolidinediones (TZDs) are a class of medicines commonly used alongside diet and exercise as therapeutic agents for the treatment of patients with type 2 diabetes mellitus, a disease in which defects in both peripheral insulin resistance and pancreatic beta cell insulin secretion coexist (Arcidiacono et al., 2015). Although used in clinical practice for many years, there is still much debate in the medical community about when TZDs should be recommended (Hiatt, Kaul & Smith, 2013). The mechanism of action of TZDs is based on their activation of peroxisome proliferator-activated receptor-γ (PPARγ), a member of the nuclear receptor superfamily of transcription factors (Janani & Ranjitha Kumari, 2015; Nanjan et al., 2018). PPARγ receptor activation by TZDs ameliorates peripheral insulin sensitivity by promoting fatty acid uptake by adipocytes and adiponectin secretion, and by suppressing the inflammatory response, which plays an important role in the development of insulin resistance. However, in spite of the beneficial effects of these molecules on insulin sensitivity and glucose homeostasis, many studies have evidenced that TZDs can lead to weight gain in several ways, including activation of adipogenesis (Colca, 2015). Also, several studies have reported that a long-term treatment with TZDs can be associated with an increased risk of cardiovascular events and cancer, thereby limiting their use in favour of other treatment options (Lewis et al., 2011; Hsiao et al., 2013; Gallagher et al., 2011; Soccio, Chen & Lazar, 2014). For example, pioglitazone, a TZD-drug in clinical use, has been associated with an increased risk for bladder cancer in people with type 2 diabetes (Tuccori et al., 2016; Piccinni et al., 2011). However, compared with pioglitazone, other TZD compounds such as rosiglitazone, were not associated with an increased risk of bladder cancer, supporting the notion that some adverse effects of TZDs are specific for each compound and not a class effect. In this regard, troglitazone, first approved by FDA in 1997 (and later withdrawn), has been reported to cause massive hepatic necrosis, while both muraglitazone and rosiglitazone have been shown to cause congestive heart failure and bone fractures (Lipscombe et al., 2007). On the other hand, a significant antiproliferative effect of TZDs on several types of cancer has been reported (Tseng, 2017; Bolden et al., 2012), together with the observation that some PPARγ agonists seem to contribute to improve the chemical sensitivity of certain cancers to standard chemotherapeutic agents (Bailey & Hollingsworth, 2013). Although clinical studies supporting TZDs as anticancer agents are limited, studies in vitro, in cell cultures, and in vivo, in animal models, well support the efficacy of TZDs in lung, breast and colon cancers (Blanquicett, Roman & Hart, 2008). In these cases, both PPARγ-dependent and -independent mechanisms which may affect apoptosis, cell cycle and differentiation have been hypothesized to explain the anti-tumor effects of these compounds.

It is evident from the literature that the TZD moiety, which is essential for the activity, is linked to the carbocyclic rings in all marketed drugs (Bruno et al., 2002). Therefore, we conceptualized that including N-heterocyclic systems like indolyl and indolinone moieties in the side chain of TZD derivatives, could result in minor side-effects of medicinal products and increase specificity for specific PPARs. In this endeavor, we synthesized indolyl/indolinone TZDs by adopting an improved (Riyaz, Naidu & Dubey, 2011) and widely applicable synthetic method.

Materials and Methods

Synthesis of TZD analogues

As reported in Fig. 1, the pharmacophore TZD [1] was prepared by treating chloroacetic acid and thiourea in the presence of the catalytic amount of HCl by refluxing in water. 1 was treated with DMF-DMA at 100 °C for 1 h furnishing 5-dimethylaminomethylene-thiazolidine-2,4-dione [2]. The structure of 2 was assigned on the basis of its IR, 1H-NMR and LC-MS data. Treatment of 2 with indole [3] in AcOH at 100 °C for 1 h resulted in the formation of 5-(1H-indol-3-ylmethylene)-thiazolidine-2,4-dione AC1 in 95% yield. The obtained compound AC1 was found to be identical in melting point and TLC with the corresponding derivative prepared in the reported method (Tseng, 2017). Using this strategy, various new indolylidine TZD derivatives (AC1-20, Fig. 2) have been synthesized and their structures have been assigned on the basis of spectral data. However, AC1-20 could also be synthesized without isolating 2, using tandem approaches, by adding indole [3] in AcOH to the reaction mixture itself. On the other hand, isatin [4] condensed with TZD [1] in the presence of methanol/KOH under refluxing condition for 30 min, leading to the formation of the TZD derivative AC21. The structure of AC21 was assigned on the basis of its IR, 1H-NMR and LC-MS data. Melting points are uncorrected and were determined in open capillary tubes in sulphuric acid bath for the crystalline products. TLC was run on silica gel G and visualization was done using iodine or UV light. Infrared (IR) spectra were recorded using Perkin Elmer 1000 instrument in KBr pellets. 1H NMR spectra were recorded in DMSO-d6 using TMS as internal standard, by using 400 MHz spectrometer. Mass spectra were recorded on Agilent LC-MS instrument.

Figure 1 Synthesis strategy of TZD compounds.

Figure 2 Structures of synthetized TZD analogues.

General procedure for preparation of 2

A mixture of TZD 1 (10 mM) and DMF-DMA (10 mM) was stirred at 100 °C for 2 h, giving rise to a colourless solid that was separated from the reaction mixture. The solid was collected by filtration, washed with hexane (10 ml) and then dried. The crude product was recrystallized from ethanol solvent to get pure 2.

General procedure for preparation of AC1-AC20

A mixture of 2 (10 mM) and indole 3 (10 mM) was refluxed for 1 h in acetic acid. At the end of this period, a light yellow coloured solid separated out from the reaction mixture, which was collected by filtration. The isolated solid was washed with hot water (10 ml) and dried. The product was recrystallized from a suitable solvent to obtain AC1.

General procedure for preparation of AC21-26

A mixture of TZD 1 (10 mM), isatin 4 (10 mM) and methanol (10 ml), followed by 40% aqueous KOH, was refluxed for 30 min. At the end of this period, solid separated out from the reaction mixture, which was collected by filtration. The isolated solid was washed with methanol (10 ml) and dried. The product was recrystallized from a suitable solvent to obtain AC21.

Spectral data of compounds are detailed in the Appendix S1.

Cell cultures

MCF-7 human breast cancer cells and PC3 human prostate cancer cells were obtained from the American Type Culture Collection (LGC Promochem, Teddington, UK). Cells were maintained in DMEM, supplemented with 2 mmol/liter L-glutamine, 50 IU/ml penicillin, 50 µg/ml streptomycin, and 10% fetal bovine serum (Iiritano et al., 2012; Gruszka, Kunert-Radek & Pawlikowski, 2005), and both viability and cell migration were assessed as described elsewhere (Gruszka, Kunert-Radek & Pawlikowski, 2005) in the presence of either home-made TZD compounds, or commercially available rosiglitazone (rosiglitazone maleate, GSK Pharmaceuticals, Worthing, UK) (Gruszka, Kunert-Radek & Pawlikowski, 2005).

MTT Assay

The MTT assay is a colorimetric assay based on the transformation of tetrazolium salt by mitochondrial succinic dehydrogenases in viable cells. By measuring mitochondrial function and metabolic activity, the MTT test provides a reproducible quantitative indication of cellular viability. In serum-free media, cultured MCF-7 and PC3 cells were tested for viability, using 12 mM MTT reagent (Sigma-aldrich, St. Louis, MO) for 2 h at 37 °C. Formazan crystals were solved with 1 mL (12-wells plate) of an organic acid solution (40% dimethylformamide, 2% glacial acetic acid, 16% sodium dodecyl sulfate, pH 4.7), in order to avoid phenol red interference. Absorbance was read after 24, 48 and 72 h treatment at 540 nm.

Wound healing

The wound healing assay is a simple method to study cell migration and cell interactions in vitro. Wound healing was performed seeding 500,000 cells/well in a 12-wells plate in order to reach confluence after 24 h. Afterwards, scretches were made using sterile 200-µL tips, then, wells were washed three times with PBS and cells were treated with RPMI medium containing the respective TZD compound (5 µM final concentration in DMSO 0.001%). RPMI plus DMSO alone (0.001%) was used as vehicle control. Photos of the wound were taken at 4, 8 and 12 h. Then, snapshots were analyzed with the ImageJ software (Arcidiacono et al., 2015) and percent of wound clousure was calculated using the following equation: %wound closure=Area t0−Area txArea t0×100

Protein extract and western blot analysis

Total and nuclear cellular extracts were prepared as previously (Brunetti, Foti & Goldfine, 1993; Brunetti et al., 1996), and Western blots were performed on total and nuclear extracts from PC3 and MCF-7 cells, after 24, 48 and 72 h treatment, as reported elsewhere (Bilotta et al., 2018). Protein extracts were resolved on sodium dodecyl sulfate polyacrylamide gel electrophoresis and electrotrasferred onto a 0.2 µm PVDF membrane (Merck-Millipore). Rabbit polyclonal antibodies anti-BCL-xL (cat. no 2762) and anti-cleaved PARP (C-PARP) (cat. no 9541) were purchased from Cell Signaling Technology. Monoclonal antibody anti-β Actin (cat. no A2228) was from Sigma-Aldrich Inc (Bianconcini et al., 2009). The resulting immunocomplexes were visualized by enhanced chemiluminescence, and densitometric slot blot analysis was performed by using the ImageJ software program.

Statistical analysis

Statistical significance was evaluated by Mann–Whitney test. p < 0.05 was considered significant. All bar graph data shown are mean ± S.E.

Molecular modelling studies

The Schrödinger Suite was adopted for computing all theoretical investigations (Software: Schrödinger, LLC, New York, NY. 2017). The 3D structure of AC-18, AC-20 and AC-22 were built using the Maestro GUI (Software: Maestro Schrödinger, LLC, New York, NY. 2017) and further submitted to LigPrep version 3.9 tool (Software: LigPrep Schrödinger, LLC, New York, NY. 2017) to take into account the most stable protomeric forms at pH 7.4. The Protein Data Bank (https://www.rcsb.org/) crystallographic entry structure 2PRG (Nolte et al., 1998) was prepared by using the Protein Preparation Wizard tool (Schrödinger, LLC, New York, NY, 2017). The molecular recognition evaluation was carried out by means of Glide software [g] in combination with Schrödinger Induced Fit docking (IFD) (Glide Schrödinger, LLC, New York, NY, 2017) protocol. The binding site of the target model was defined by means of a regular grid box of about 64,000 Å centered on the co-crystallized ligand. Docking simulations were computed using Glide ligand flexible algorithm version 6.7 at standard precision (SP).

Results

Inhibition of cell viability in MCF-7 and PC3 cells

All the synthetic TZD-based compounds (from AC1 to AC26) were initially screened for their in vitro effect on cell viability, using MTT assays in MCF7 and PC3 cells, that underwent treatment with 5 µM TZD analogues over a period from 24 to 72 h of exposure (Figs. 3A and 3B). As observed both in MCF-7 cells and PC3 cells, not all compounds investigated produced the same effect on cell viability, as compared with untreated control cells. In fact, whereas some of them had no influence on this parameter at all time points of the 72-h period in both cell lines, a decrease of cell viability (with slight difference among MCF-7 and PC3 cells) was observed with AC11, AC12, AC14, AC15, AC17, AC18, AC20, AC21, AC22, AC23, AC24, AC25, and AC26 (Figs. 3A and 3B).

Figure 3 Effects of TZD compounds (AC1–AC26) on cell viability.

MTT assays were performed in MCF7 (A) and PC3 (B) cells as reported in the Materials and Methods section. Optical density (OD) was measured at 540 nm, after 24, 48 and 72 h drug treatment. Results are the mean ± S.E. of triplicates from three independent experiments. *p < 0.05, **p < 0.01 relative to untreated control cells (C = 1), which received dosing vehicle alone (0.001% DMSO). The selected AC18, AC20 and AC22 TZD analogues, which were the most active in the inhibition of cell viability in MTT assays, are shown.

Next, both cell types were incubated with increasing concentrations (1 to 33.3 µM) of each of the twenty-six compounds individually to calculate IC50 values (Table 1). Among all the compounds evaluated, compounds AC18, AC20 and AC22 were found to be the most promising, showing a 50% reduction of cell viability in both PC3 and MCF-7 cells after 48 h exposure to 5 µM concentration of each of the three drugs. However, whereas the inhibition of cell viability in MCF-7 cells was already significant at 48 h with all three TZD analogues, no significant inhibition of cell viability by these compounds was observed before 72 h incubation in PC3 cells (Figs. 3A and 3B), thus suggesting some component of cell specificity amongst these synthetic drugs.

Table 1 IC50 values of all the 26 compounds tested with MTT assay.

Dose-response and time-course experiments were performed by exposing MCF-7 and PC3 cells to increasing doses (1.0, 3.3, 5.0, 10.0, 33.3 µM) of each TZD compound for 24, 48 and 72 h. IC50 values are expressed as the concentration of each compound required to produce 50% inhibition of cell viability, in relation to time of exposure to the compound. NR, not responsive.

Compound	MCF-7	PC3	Compound	MCF-7	PC3	
1	NR	NR	14	33.3 µM (72 h)	33.3 µM (72 h)	
2	NR	NR	15	NR	NR	
3	NR	NR	16	NR	NR	
4	NR	NR	17	33.3 µM (72 h)	33.3 µM (72 h)	
5	NR	NR	18	5 µM (48 h)	5 µM (48 h)	
6	NR	NR	19	NR	NR	
7	NR	NR	20	5 µM (48 h)	5 µM (48 h)	
8	NR	NR	21	33.3 µM (72 h)	33.3 µM (72 h)	
9	NR	NR	22	5 µM (48 h)	5 µM (48 h)	
10	NR	NR	23	33.3 µM (72 h)	33.3 µM (72 h)	
11	NR	NR	24	NR	NR	
12	33.3 µM (72 h)	33.3 µM (72 h)	25	33.3 µM (72 h)	33.3 µM (72 h)	
13	NR	NR	26	33.3 µM (72 h)	33.3 µM (72 h)	

To further evaluate the effect of AC18, AC20 and AC22 on cellular viability, we performed MTT assays, in which the effect of each of these compounds on cell proliferation was compared to that of untreated control cells, or cells treated with either the negative AC1 TZD compound, or 5 µM rosiglitazone, a potent oral antidiabetic drug, whose inhibitory role on cell viability has been reported in vitro, in rat prolactin-secreting pituitary tumor cells (Gruszka, Kunert-Radek & Pawlikowski, 2005). As shown in Figs. 4A and 4B, treatment of both MCF-7 and PC3 cells with 5 µM of either AC18, AC20 or AC22 induced a significant decrease of cell viability as compared with control cells, which, in some instances, was even more evident than that obtained with the same dose of rosiglitazone.

Figure 4 Comparison between Rosiglitazone, AC18, AC20 and AC22 on cell viability.

MTT assays were performed in MCF7 (A) and PC3 (B) cells, either untreated or treated as indicated. OD was measured at 540 nm, after 24, 48 and 72 h drug exposure. Results are the mean ± S.E. of triplicates from three independent experiments. *p < 0.05, **p < 0.01, ***p < 0.001 relative to either untreated control cells (C = 1), or cells treated with the non-effective AC1 compound. Representative Western blots of BCL-xL and C-PARP from cell extracts of untreated and treated MCF-7 and PC3 cells are shown in the autoradiograms, together with densitometric results of BCL-xL and C-PARP proteins over b-Actin. All protein samples were processed at the same time, under the same experimental conditions, PVDF membranes simultaneously exposed to ECL detection reagent, and immunocomplexes visualized by enhanced chemiluminescence in the dark for 1 min. Autoradiograms were generated by positioning the membranes in the same X-ray cassette and thus exposed to the same film, in the same exposure conditions. Rosi, rosiglitazone.

To further support the antiproliferative potential of AC18, AC20 and AC22, Western blot analyses of BCL-xL were carried out in cell lysates from MCF-7 and PC3 cells. BCL-xL is an important anti-apoptotic member of the BCL-2 protein family and a potent regulator of cell death. While overexpression of BCL-xL in PC3 and MCF-7 cells has been reported to contribute to the apoptosis-resistant phenotype of these cells in response to products with antitumor activity (Li et al., 2001; Okamoto, Obeid & Hannun, 2002), the reduction of Bcl-xL levels rendered these cells more sensitive to the drugs (Yu et al., 2014; Lin, Li & Zhang, 2016), thus reducing cell survival. As shown in Figs. 4A and 4B, BCL-xL protein levels were reduced in both MCF-7 and PC3 cells treated with the AC22 TZD compound, and this reduction paralleled the decrease in cell viability (as monitored by the MTT assay). Additionally, as shown in Fig. 4B, treatment of PC3 cells with either AC18 or AC22 showed a better inhibition of cell viability compared with cells treated with rosiglitazone, and also in this case a parallelism was observed between cell viability and BCL-xL protein expression. Therefore, it is tempting to hypothesize that the antiproliferative effect of these TZD compounds, including rosiglitazone, on MCF-7 breast cancer cells and PC3 prostate cancer cells could be mediated, at least in part, by the reduction in BCL-xL expression. These data were in agreement with the analysis of C-PARP, a marker of apoptotic response (Kraus, 2008), indicating that C-PARP expression was higher in MCF-7 and PC3 cells treated with the AC22 TZD compound than in rosiglitazone-treated cells.

Effects of AC18, AC20 and AC22 on cell migration

The effects of AC18, AC20 and AC22 on cell migration were studied as well. To this end, wound healing assays were performed in vitro with both MCF-7 and PC3 cells, either untreated or treated with the synthetic TZD agonists, at 4 h time points (up to 12 h). As shown in Figs. 5A and 5B, a significant inhibition of cell migration was observed in both MCF-7 and PC3 cells, in the presence of compounds AC18, AC20 and AC22, as demonstrated by the decreased migratory response of cells in response to wound healing. Also in this case, as for the MTT test, the effect of all three compounds on cell migration was comparable or even superior than that of rosiglitazone (Figs. 5A and 5B), thereby indicating that these novel compounds may indeed represent a new potential class of antiproliferative agents.

Figure 5 Inhibition of cell migration.

Wound healing assays were carried out in MCF7 (A) and PC3 (B) cells, using 200 µL pipette tips to scratch confluent cells on the base of a 12-well plate. Wound healing (% wound closure) was measured and analyzed with the NIH ImageJ software in both cell types, after 4, 8 and 12 h incubation with the compounds. Results are the mean ± SE of triplicates from three independent experiments. *p < 0.05, **p < 0.01, ***p < 0.001 relative to untreated (control) cells.

Theoretical binding affinity of compounds AC18, AC20, AC22 in the PPARγ active site in comparison to rosiglitazone

In order to understand the molecular interactions of AC18, AC20 and AC22 into the PPARγ active site, ligand-target recognition studies were performed using the Glide Suite Docking package (Glide Schrödinger, LLC, New York, NY) and refined with the Schrödinger Induced Fit Docking (IFD) protocol (Software: Induced Fit Docking protocol, Schrödinger, LLC, New York, NY) (Table S1, Appendix S1). We selected the rosiglitazone/PPARγ crystal structure (PDB ID: 2PRG) (Nolte et al., 1998) to carry out the docking simulations. We validated our protocol by calculating the Root Mean Square Deviation (RMSD) on the ligand Cα, thus demonstrating that the standard precision (SP) protocol was able to well reproduce the co-crystallized pose (rosiglitazone re-docking RMSD value: 1.38 Å). The receptor grid center was specified from the bound ligand and the energy window for ligand conformational sampling was 2.5 kcal/mol. In order to investigate the binding domain flexibility, the residues within 5.0 Å from the ligand poses were refined using the Prime molecular dynamics module (Prime Schrödinger, LLC, New York, NY, 2017). A maximum of 20 poses was generated. The docking results clearly indicated that all compounds are able to geometrically reproduce rosiglitazone binding mode (Fig. 6). Indeed, by analyzing the docking poses of all compounds, we observed that the ligands were well accomodated into the active site of PPARγ by establishing hydrogen bonds (HBs) between their carbonyl groups of the thiazolidine moiety with H343, Y473 and H449 residues. Furthermore, the two water molecules located into the binding site are involved in an indirect HB network between the compounds and the binding pocket residues. However, it is interesting to note that for AC20 and AC22 the HB with R288 is lost, while it is maintained by AC18. AC18 and AC20 share the same methoxyl group at position 5 of their indole ring, while in AC22 it is substituted with a fluorine atom. Probably the ethyl group on the amine of the thiazolidine ring, absent in AC18 and AC22 compounds, could be responsible for a different geometry of the molecule.

Figure 6 In silico Induced Fit Docking of TZD compounds into the PPARγ binding pocket.

Rosiglitazone (A) is shown as light green carbon sticks, while AC18 (B) AC20 (C) and AC22 (D) are shown as green, white and cyan ball-and-sticks, respectively. The residues located within 4 Å from the bound ligand are displayed (gray sticks) and labeled. Hydrogen bonds, bad and ugly contacts between the ligands and the enzyme residues or water molecules are depicted with yellow, orange and red dashed arrows, respectively.

Discussion

Insulin resistance is a hallmark of obesity and type 2 diabetes (Pullinger et al., 2014; Greco et al., 2014; Chiefari et al., 2018), and both these disorders have an increased risk to develop some types of tumours, including breast, liver, colon, and pancreatic cancers (Arcidiacono et al., 2012; Talarico et al., 2016). Many clinical trials have also documented that patients treated with TZDs, while improving glycemic control and in spite of potentially drug-related adverse effects, are less predisposed to these kind of cancers (Janani & Ranjitha Kumari, 2015). The observation that some antidiabetic agents such as TZDs (Janani & Ranjitha Kumari, 2015; Tseng, 2017; Bolden et al., 2012; Bailey & Hollingsworth, 2013; Costa et al., 2008), as well as metformin (Anisimov, 2015; Salani et al., 2014; Lau et al., 2014), may exert an antiproliferative effect on many cell types has triggered an intense research effort in the last years. Activation of PPARγ by TZDs, by inducing apoptosis, growth arrest and cell differentiation in cancer cells, has been proposed as an explanation for the anticancer activity of these compounds (Blanquicett, Roman & Hart, 2008), together with PPARγ-independent mechanisms involving the proteosomal degradation of cyclins D1 and D3 (Lu et al., 2005), as well as the upregulation of PTEN/AMPK and the downregulation of the Akt/mTOR signaling pathways (Han & Roman, 2006). Further mechanisms of tumor suppression by TZDs may involve the inhibition of important target genes, such as the vascular endothelial growth factor, VEGF, gene (Yang et al., 2005), the PGE2 receptor gene (Lebovic et al., 2013), and the insulin receptor gene (Costa et al., 2008).

Nevertheless, our knowledge and understanding of how TZDs may exert anti-tumor effects still remain not fully understood. This is made all the more difficult by the fact that non-univocal results still exist on the antiproliferative activities of these drugs, either in in vitro studies or in clinical investigations. In fact, whereas a strong antiproliferative effect was seen for TZDs in several studies, no significant inhibition of cell proliferation of these compounds was observed in a number of other similar investigations from different research groups (Lewis et al., 2011; Hsiao et al., 2013; Tuccori et al., 2016). Conflicting results, in this direction, have been reported also by meta-analysis of clinical trials, in which no association was found between the use of TZDs and breast cancer risk among diabetic women (Du et al., 2018), whereas a protective effect of TZDs was associated with colorectal cancer in patients with diabetes (Liu et al., 2018). Based on these considerations, there is a need to better understand the molecular effects of TZDs and to develop safer and more effective synthetic ligands of PPARs. Herein, we explored the antiproliferative effects of a series of newly synthesized TZD compounds, in which the carbocyclic ring was replaced by the N-heterocyclic ring in an attempt to reduce their toxicity while retaining their antiproliferative abilities at the same time. The replacement of the 2-pyridinylamino terminal scaffold of rosiglitazone with the indolo ring prevents the potential formation of N-nitroso metabolites able to exert toxicity in vivo (WHO, 1978). Conversely, the indolo ring is safer as typical scaffold of TRP sidechain. Therefore, by adopting a simple and tunable synthetic strategy, twenty-six different TZD compounds were synthesized in moderate to high yields. Among them, only three, designated AC18, AC20 and AC22, showed a significant antiproliferative effect in two different human neoplastic cell lines, MCF-7 breast cancer cells and PC3 prostate cancer cells, as demonstrated by cell viability assays, further supporting the notion that different TZD compounds may display different effects on cell viability. In agreement with previous observations with rosiglitazone (Wang et al., 2016), as revealed by Western blot analysis, part of this antiproliferative effect was supported by apoptosis studies in both cell models, showing downregulation of BCL-xL and upregulation of C-PARP following drug treatment. As a next step towards their characterization, these same TZD molecules were also effective in inhibiting cell migration and cell-to cell interaction, as shown by wound healing assays. Furthermore, as the same molecule may present different efficacy and potency on different cancer cell lines, we showed that AC18 and AC22 were both more effective in MCF-7 and in PC3 cells than rosiglitazone, whereas AC20 was more powerful than rosiglitazone in MCF-7, but not in PC3 cells. This can be partially due to the poor differentiation grade of PC3 cells (Tai et al., 2011) compared with MCF-7 cells. Also, PC3 cells are derived from a prostate cancer metastasis, while MCF-7 cell strain retains many characteristics of the mammal epithelium cells, such as the presence of estrogen receptors (Holliday & Speirs, 2011). However, since the antiproliferative activity of these compounds has been assessed in vitro, we can only speculate about their safety in vivo, and more studies are necessary to clarify this issue. In particular, further investigation in animal models would confirm both efficacy and safety of the compounds herein proposed. Notably, the more efficacious activity of AC18, AC20 and AC22 compounds, as compared to rosiglitazone in an in vitro biological setting, was supported in in silico studies. This, in our opinion, represents a strength of the present work, together with the fact that the present study is one of the few reports in the literature analyzing anticancer properties of TZDs by wound healing assay. Conversely, as limitation of this study, we should mention that we could not investigate the expression of specific proliferation markers. Overall, the interest of TZDs as adjuvant therapy in cancer treatment is still ongoing, and this is further supported by recent data indicating that also noncanonical PPAR γ agonists may modulate cancer cell sensitivity to chemotherapy for therapeutic gain (Khandekar et al., 2018).

Conclusions

Herein, we propose a new set of three TZD analogues as potential anticancer agents. Although more effort is needed to understand their efficacy and safety, we believe that these compounds have the potential to be further developed as a novel adjuvant tool for anticancer treatments.

Supplemental Information

Appendix 1 Supplementary Appendix

Spectral data of compounds and Table S1.

Click here for additional data file.

Data S1 Raw data

Click here for additional data file.

Additional Information and Declarations

Competing Interests

Author Contributions

Data Availability

The authors declare there are no competing interests.

Domenica M. Corigliano performed the experiments, analyzed the data, prepared figures and/or tables, authored or reviewed drafts of the paper, approved the final draft.

Riyaz Syed analyzed the data, contributed reagents/materials/analysis tools, prepared figures and/or tables, authored or reviewed drafts of the paper, approved the final draft.

Sebastiano Messineo conceived and designed the experiments, performed the experiments, prepared figures and/or tables, authored or reviewed drafts of the paper, approved the final draft.

Antonio Lupia performed the experiments, prepared figures and/or tables, authored or reviewed drafts of the paper, approved the final draft.

Rahul Patel, Chittireddy Venkata Ramana Reddy, Pramod K. Dubey, Giovambattista De Sarro and Adisherla Indrasena analyzed the data, contributed reagents/materials/analysis tools, authored or reviewed drafts of the paper, approved the final draft.

Carmela Colica and Stefano Alcaro analyzed the data, authored or reviewed drafts of the paper, approved the final draft.

Rosario Amato performed the experiments, authored or reviewed drafts of the paper, approved the final draft.

Antonio Brunetti conceived and designed the experiments, analyzed the data, contributed reagents/materials/analysis tools, prepared figures and/or tables, authored or reviewed drafts of the paper, approved the final draft.

The following information was supplied regarding data availability:

The raw data are provided in a Supplemental File.

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
