# Peer review of "Indole and 2,4-Thiazolidinedione conjugates as potential anticancer modulators"

_PeerJ, doi:10.7717/peerj.5386_

## Round 0.1 · original submission · Major Revisions

Mislabeling and misreferring of figures should be corrected. Experimental methods and new assays should be properly written and performed. Expand discussion section. Statistical analysis section should be properly provided in the method section. Controls should be provided for each experiment as well as migration assays should also be provided. Please see detail reviewers comments.

Reviewer 1 ·

Basic reporting

1. Repeated mislabeling and misreferring of figures in both figure legend and result sections. For example, figure legend-4 refers to the figure-1 for the MTT assay scheme but figure-1 is the synthesis strategy of the TZD compounds. Line-222; Figure 4A and 4B are referred for the cell migration assay but the figure-4 data shows cell viability.
2. Figure-4 and Figure-5 (MTT assay) are the subset of data in figure-3. There is no reason to project the same data repeatedly in different figures.
3. Line 189-192; the table 1 shows IC50 of all the three compounds (18, 20, and 22) are 5μM at 48h. However, the statement made in this line is not supported by the data in table 1. Moreover, the figure referred for the statement (Figure2) has no relevance with the statement. Please explain.

Experimental design

4. How do you assay the IC50? Is it different from MTT assay? If not, I suggest to combined in a single paragraph under the heading “Inhibition of cell viability in MCF-7 and PC3 cells” without “IC50” or “MTT” as heading.
5. Though the treatment of the cells with the TZD compounds increases BCL-xL in temporal manner (gradually increased from 24h to 72h) but the authors claims other way around. The increase BCL-xL is thought to act as anti-apoptotic. Please compare 24h, 48h and 72h of each compound, the BCL-xL is either increases or unaltered. Please explain.
6. I suggest assaying for other proapoptotic marker in the treated samples, for example Caspase 3 or PARP cleavage.
7. The endogenous loading control is missing in the BCL-xL western blots.
8. I suggest the authors to assay the binding affinity of compounds 18, 20, and 22 for the PPARs in comparison to Rosiglitazone.

Validity of the findings

9. Line 188; the statement claims IC50 5μM at 72h whereas data provided in table-1 shows IC50 5μM at 48h. Please clarify.
10. Introduction should include more details background on different genera of TZD and their specific drawbacks and the knowledge gap should be established clearly.
11. Please explain why RPMI media was used for the wound healing assay (line 160) while cells were maintained in DMEM media.

Additional comments

12. Please provide literature support for concept why indolyl and indolinone sidechain may decrease side effect of TZD (line-72-74).
13. Please provide the buffer composition used for total cell extract preparation and catalog number of the antibodies used.
14. Move down line 97-101 to the end of the “Synthesis of TZD analogues” section.

The manuscript entitled “Indole and 2,4-thiazolidinedione conjugates as potential modulators of cellular apoptosis and proliferation” by Messineo et. al. has tried to address an important clinical problem by a synthetic chemistry approach. I commend the authors for their extensive effort for the synthesis of TZD derivatives. I found the conjugation of heterocyclic moieties to the TZD derivatives synthesis are compelling. Though, several efforts have been reported in the literatures (reviewed by M.J.Nanjan et.al. Bioinorganic Chemistry,2018) for the synthesis of TZD derivatives in the past to reduce side effect and increase specificity for specific PPARs. I think the authors fail to build a strong rationale in support to their specific synthesis approach why they specifically synthesize indolyl/indolinone TZDs, thought indolyl/indolinone conjugation thought to reduce pharmacological side effects. Overall manuscript is poorly written with redundant experimental data and several mislabeling. The authors repeatedly presented same data at multiple figures. At the present stage of the manuscript, I don’t recommend considering for the publication in PeerJ unless a major revision address all the concerns.

·

Basic reporting

The manuscript #24396, entitled “Indole and 2,4-thiazolidinedione conjugates as potential modulators of cellular apoptosis and proliferation” by Messineo et al. describes the anti-proliferative and anti-migratory capacity of three novel TZD derivates, suggesting their putative utility as anti-tumor compounds.

This is an interesting study expanding the putative utility of TZD-derived compounds and providing relevant information regarding novel anti-tumor candidates. Only few comments are raised:

1) The title should be adapted in that it does not accurately reflect the results presented herein. No direct assessment of apoptosis capacity is demonstrated and this should be eliminated from the tittle.
2) Abstract, line 36: Should it say migration instead of proliferation?
3) The discussion section can be expanded substantially, incorporating the discussion of additional previous results.

Experimental design

1) In the western-blot section of the material and methods, more extended information should be provided in terms of methods used for protein extraction, blotting procedures and data analyses.
2) The statistical analysis description is missing in the material and methods section. It should be provided.
3) It is odds that the authors used apoptosis markers to demonstrated anti-proliferative effects, in that these effects are not necessarily mediated by apoptosis induction. The authors should want to separate both sections by 1) adding some proliferation marker and 2) demonstrating induction of apoptosis. This could greatly increase the impact and significance of the study

Validity of the findings

No comment

Reviewer 3 ·

Basic reporting

No comment

Experimental design

In the manuscript by Messineo et al, the authors detail the rationale for designing analogues of Thiozolidinediones (TZD). They have done a commendable job in designing 26 derivative and testing them for anti-proliferative effects in human breast and prostate cancer cell lines by MTT assay. However, several key controls were missing in the subsequent evaluation of apoptosis-western blots were missing loading controls and migration assays were missing zero hour controls.

Validity of the findings

The data lacks important controls (see below). Based on this the conclusions are over stated claiming them to have promising potential in treating breast and prostate cancer.

Additional comments

In the manuscript by Messineo et al, the authors detail the rationale for designing analogues of Thiozolidinediones (TZD). They have done a commendable job in designing 26 derivative and testing them for anti-proliferative effects in human breast and prostate cancer cell lines by MTT assay. Based on the IC50 curves, the authors prioritized AC18, AC20 and AC22 to carry out cell migration and evaluation of anti-apoptotic markers. Listed below are some of the major concerns regarding this manuscript:
1. The MMT assays show that AC 19, 25 ,26 show significant anti proliferative effects with effects as early as 24h but these did not make the cutoff based on IC50 values and these discrepancies were not addressed.
2. For the blots demonstrating changes in Bcl-xL, the western blots lacked loading controls and therefore the results described by the authors could not be validated. Further, for apoptosis induction, the direct analysis of cleaved PARP and cleaved caspase 3 are considered gold standard assays, which were not evaluated in this manuscript. Alternatively, the authors could potentially use propidium iodide based staining to confirm cell death by apoptosis.
3. For the analysis on cell migration by the wound healing assay, the critical 0 hour control images were missing. This result is necessary for the evaluation of width of the scratch at time 0 and consistency of this width between the different treatment conditions. In the absence of this control, it is not a well-controlled experiment to draw conclusions about the different responses.

---

## Round 0.2 · Minor Revisions

Please fix western blot data issue for cell death and other minor issues for reconsideration.

Reviewer 1 ·

Basic reporting

The authors have improved overall quality of the manuscript significantly. I would recommend the paper to be accepted for publication in PeerJ.

Experimental design

No comment

Validity of the findings

No comment

·

Basic reporting

No comment

Experimental design

No comment

Validity of the findings

No comment

Additional comments

The authors have appropriatelly addressed all my comments.

Reviewer 3 ·

Basic reporting

The manuscript is clearly written and easy to follow. The figure labeling has been corrected.

Experimental design

The authors have addressed the concern of loading controls for western blots. However,
for the key comparison of Bcl-xl levels between control and test samples (AC18,20,22), the blots should not be separated and be analysed in the same exposure conditions in Fig 4A and B.

In the PARP blot, the trend in both C-PARP and b-Actin is the same, suggesting no increase in the cleaved form of PARP with Rosi or AC22 in Fig 4A.

The b-actin blot in Fig 4B is over exposed and therefore making the results hard to interpret.

Validity of the findings

The authors have addressed the other concerns made in
1. rationale for picking AC 18, AC20 and AC22

3. addition of 0h controls in Figure 5.

Additional comments

The western blot data for cell death is not convincingly supporting the claims made in the paper.

---

## Round 0.3 · accepted · Accept

Authors have addressed remaining issues and manuscript is ready for publication.